# Report of 16 Years of the BCG Vaccine under the Expanded Program on Immunizations in Mexico (2006–2021)

**DOI:** 10.3390/vaccines11020337

**Published:** 2023-02-02

**Authors:** Rodrigo Romero-Feregrino, Raul Romero-Cabello, Mario Alfredo Rodríguez-León, Raúl Romero-Feregrino, Berenice Muñoz-Cordero, Julieta Isabel Aguilar-Feregrino

**Affiliations:** 1Asociación Mexicana de Vacunología, Mexico City 06760, Mexico; 2Instituto Para el Desarrollo Integral de la Salud (IDISA), Mexico City 06700, Mexico; 3Employer Sector CONCAMIN, Technical Council, Instituto Mexicano del Seguro Social (IMSS), Mexico City 06600, Mexico; 4Saint Luke School of Medicine, Mexico City 11000, Mexico; 5Department of Infectology, Hospital General de México, Mexico City 06720, Mexico; 6Department of Microbiology and Parasitology, Faculty of Medicine, Universidad Nacional Autónoma de México (UNAM), Mexico City 04360, Mexico; 7Medical Surgeon Career, Faculty of Higher Studies Zaragoza, Universidad Nacional Autónoma de México (UNAM), Mexico City 09230, Mexico; 8Tecnológico de Monterrey, Mexico City 14380, Mexico

**Keywords:** Bacillus Calmette and Guérin, BCG vaccine, low coverage, vaccine purchase, Universal Vaccination Program, Mexico

## Abstract

Background: In recent years in Mexico, a decreased Bacillus Calmette and Guérin (BCG) coverage has been observed concomitantly with new cases of tuberculosis. Material and Methods: This study is a descriptive and analytical evaluation regarding both BCG vaccine acquisition and coverage as reported by official sources over a 16-year period (2006–2021). Results: We found that vaccine acquisition, dose application and coverage are highly variable each year. Coverage is 90% or higher, except for the 2017–2020 period. Discussion: According to our calculations, between 3,917,616 and 4,961,868 individuals did not receive the BCG vaccine. Coverage was lower than 90% during the last 4 years, whereas this value decreased to 21% in 2020. Except for the last 5 years, the amount of acquired doses surpassed the demand thus causing a considerable vaccine wastage. Conclusions: BCG vaccine coverage is low and many individuals remain unprotected. The access to this vaccine is difficult and the number of newly reported cases of tuberculosis have increased during the last years. Thus, it is necessary to establish vaccination campaigns aimed protect the population and also to deploy a nominal system to control coverage, acquisitions, and target population.

## 1. Introduction

The Bacillus Calmette and Güérin (BCG) vaccines are among the oldest vaccines and were first used in humans in 1921. BCG is a live attenuated bacterial vaccine derived from *Mycobacterium bovis* that was originally isolated in 1902 from a tuberculous cow. World Health Organization (WHO) recommends that any BCG vaccine used in immunization programs should comply with their standards. Currently, BCG is the only TB vaccine available. Whereas BCG has shown a significant effectiveness when applied to several populations, its protective effect for several age groups has not been consistent against most TB forms, e.g., a 44–99% protection has been observed against neonate pulmonary tuberculosis, 85–90% against severe TB, 73% against TB meningitis, and 77% against miliary TB. Consequently, several new TB vaccine candidates are currently under development, and some of them have reached the advanced clinical trials. These new vaccines are designed to be used as booster vaccination after neonatal BCG vaccination [1].

In those countries or settings characterized by a high TB incidence and/or high leprosy burden, for prevention purposes, a single dose of the BCG vaccine should be applied to every healthy neonate right after birth. If the application of the BCG vaccine is not possible in the latter conditions, it should occur at the earliest opportunity thereafter without delay in order to protect the infants before a possible exposure to an infection occurs [1].

In terms of coverage, a decrease has been observed in recent years in Mexico: from 96% in 2018 to 76% in 2019, 33% in 2020, and 86.6% in 2022, as reported by the Pan American Health Organization (PAHO) and the National Health and Nutrition Survey 2021 on Covid-19. Similarly, a supply shortage of several vaccines has been reported in 2019, including BCG [2,3,4].

In Mexico, the BCG vaccine was first applied in 1951, although an overall vaccination program envisaging children above 5 years old was not executed until 1991 when a government planning involving an inter-institutional logistics was established under the name “Universal Vaccination Program”. After executing this program, one disease was effectively eradicated and others were controlled, whereas a ≥90% vaccination coverage was achieved [4,5]. However, in recent years such coverage has decreased, as reflected by the increased incidence of some vaccine-preventable diseases. Tuberculosis, either as a primary infection or as facilitated by immunocompromising diseases, is of vital importance because it has re-emerged as a public health problem [1,6,7,8,9,10,11,12,13].

The newly diagnosed cases are officially registered in Mexico every year, as all tuberculous meningitis and respiratory tuberculosis cases occurring in the country must be reported. The official figures are the following (age was not indicated) (Table 1):

A decreased vaccine coverage has been observed in recent years (including the BCG vaccine), particularly during the COVID-19 pandemic. The estimated global coverage in America was 88% in 2019, and it subsequently decreased to 83% the same year when compared to a 92% coverage estimated the previous year [2].

BCG vaccination of infants right after birth or at the possible earliest moment, is a key component of the ‘pillar 1′ tier envisaged by the ‘End TB Strategy’. The latter is a WHO program aimed to decrease TB deaths by 95% in 2035 as compared to the 2015 rates. It has been estimated that a high global coverage (90%) and the widespread use of BCG in routine children vaccination programs may prevent over 115,000 TB deaths per birth cohort within their first 15 years of life [1].

The review of the financial information regarding children BCG vaccination suggests that it is cost-effective in developing countries and in settings where TB incidence is >20/100,000 population or where >5/100,000 smear-positive cases occur per year [1].

Based on the data showing a gradual decrease of vaccination coverage concomitant with an increased number of diagnosed cases in recent years, the reports indicating shortages of vaccine supply, in addition to our knowledge on vaccinology, vaccine acquisition, and our access to information regarding the Mexican health system, we considered that it is necessary to study the official data on acquisitions, application, and coverage, in order to ascertain the actual situation of our country in terms of BCG vaccination. Our aim is to understand the underlying causes and to subsequently propose solutions to address the issues related to vaccination. The following questions arise: which situations contribute to this decreased vaccination coverage?, what are the causes?, and how can we improve this situation?

To answer to these questions and to address other related issues, this report reviews and analyzes the information available on the BCG vaccine in Mexico during the previous 16 years as obtained by the major public health institutions. The obtained data on acquisitions, application, and coverage, will comprise the basis to generate proposals aimed to urgently improve the tuberculosis vaccination program.

In this report, we expect to find a low agreement between data reported by public health institutions and our own calculations concerning the amount of acquired vaccines, the number of applied doses, and coverage values.

## 2. Material and Methods

This study is of descriptive and analytical nature. It is based on the information regarding BCG vaccination in Mexico available in official sources within a 16-year period: 2006–2021.

We focused on vaccine acquisition and coverage as reported by the three major government health institutions in Mexico. We analyzed and compared data on the amount of purchased doses by each institution, as well as the information concerning the target population and the number of applied doses of the BCG vaccine that were acquired within a year.

The major government health institutions in Mexico are: the Mexican Social Security Institute (IMSS, acronym in Spanish), the Security and Social Services Institute for State Workers (ISSSTE), and the Ministry of Health (SSA). The combined effort of these three institutions attend to 98% of the medical care needs of a country that has a population surpassing 128 million inhabitants, including 2.2 million births per year, as reported in 2020 [14,15,16,17,18,19,20].

We established models to compare the number of acquired vaccine doses, their applications, and the overall coverage.

The accuracy and reliability of the study depends on having an abundant amount of data from official sources regarding the amount of acquired doses and their application as reported by the previously mentioned institutions within the 2006–2021 period. This information will be used to establish suitable theoretical models in order to identify a trend or to provide an adequate explanation of the results. This was also important to obtain clear conclusions aimed to make proposals and to suggest actions for improvement purposes.

This study was conducted based on the following steps:

### 2.1. Information

We established the type of data required to conduct the study, i.e., beneficiaries per institution, vaccine description, number of acquired doses, coverage by either ISSSTE, IMSS, or Ministry of Health (SSA), and coverage reported to the Pan-American Health Organization (PAHO). All the information was queried and obtained from the following databases: Compranet [21], IMSS purchase [22] and PAHO [23].The access to this information was requested to the National Institute for Transparency, Access to Information, and Protection of Personal Data (INAI, acronym in Spanish) [24,25,26,27,28,29,30,31,32,33,34,35,36,37,38,39,40,41,42,43,44,45,46].

### 2.2. Analysis

After the above information was obtained, we conducted the following analyses.

#### 2.2.1. Number of Acquired Doses

We established models to compare the amount of acquired doses with their actual number of doses, and their application. The following terms were considered for data interpretation:Theoretical target population: the number of individuals that should be vaccinated based on the population considered by the respective institution and the indications for the vaccine.Percentage variation of annual acquisitions: percentage variation of annual purchases after comparing a particular year with the next one.%PUR: percentage of acquired vaccines regarding the theoretical target population.%APP: percentage of applied vaccines regarding those acquired by the respective institution.%COV: percentage of applied vaccines regarding the theoretical target population.

The theoretical target population is a calculation made to identify the necessary number of vaccines for a particular target population based on data obtained from the CONAPO database [15,16,17], and also on indications for the vaccine according to Mexico’s national vaccination scheme. The overall number of vaccines was increased by 10% to compensate for general losses.

The theoretical target population by institution is a calculation that we made of the necessary amount of vaccine doses per individual institution, i.e., IMSS, ISSSTE, and SSA. The following data were considered: the theoretical amount of required doses and the respective percentage for a particular user population as reported by each institution annually (Table 2). Based on these data, we calculated the target population percentage per institution [14,15,16,17,18,19,20].

The percentage variation for the analysis of annual purchases and the amount acquired annually per institution were evaluated using the following expression:(acquired amount − amount acquired the previous year) ∗ 100amount acquired the previous year(1)


#### 2.2.2. Acquisition, Application, and Coverage

A comparative report was made regarding the acquisition percentage for a particular target population as well as the percentage of vaccine application:%PUR = (amount of acquired doses/theoretical target population) ∗ 100(2)
%APP = (number of applied doses/amount of acquired doses) ∗ 100(3)
%COV = (number of applied doses/theoretical target population) ∗ 100(4)

The data were processed using the Microsoft^®^ Excel software and dynamic tables and graphs were prepared with different data combinations. Only representative data were shown.

Finally, we compared the %COV with the coverage value as reported by PAHO [3] and IMSS.

## 3. Results

The amount of data on the acquired amount of BCG vaccines by all 3 institutions during the 2006–2021 period was: 14 for SSA, 14 for ISSSTE, and 9 for IMSS, from an overall of 16 queried data. Most of the information was successfully retrieved, except for the IMSS 2006–2011 period, the ISSSTE 2015–2021 period, and the SSA data from 2019 and 2021. The former data were not obtained because either it was not reported or no acquisitions were made in those years.

### 3.1. Acquired Doses

Figure 1 shows the identified amounts of BCG doses. A comparison was made between the annual acquisitions by each institution with the overall amount (Figure 1).

Annual acquisitions are highly variable regarding the purchased number of vaccines (Figure 2). The average, maximum and minimum values per institution were the following, respectively: –5.35%, 16.71%, and –24.52% for IMSS; 198.70%, 2313.90%, and –77.70% for ISSSTE; –0.72%, 65.19%, and –49.70% for SSA, whereas the total amount of doses were 3.64%, 137.34%, and –62.30%.

For instance, variation was 2313.90% for ISSSTE in 2008, thus the acquired doses increased from 100,000 to 2,413,900 doses in that year. Variation was –42.43% for SSA in 2010, and this implies that the acquired doses decreased from 4.2 to 2.1 million.

### 3.2. Coverage

Table 3 and Figure 3 show the acquisition percentage regarding the theoretical target population and the percentage of vaccine application.

Figure 3 shows a coverage evaluation per institution based on the target population and vaccine application as well as the overall value (all 3 institutions). Most of the coverage by ISSSTE and IMSS is low, whereas that provided by SSA surpasses 100%, according to the theoretical population. Furthermore, all three institutions provide a 90% coverage or more, except for the2017–2020 period.

Table 4 show the results sorted per institution and the overall number of unvaccinated individuals each year. This value was obtained by subtracting the number of applied vaccines from the theoretical target population size. Negative numbers represent unvaccinated individuals whereas positive numbers are those vaccinated regarding the proposed theorical target. The calculated %COV value is also shown.

The data from Table 4 was recalculated: the theoretical target population size in 2006 was subtracted from the total value. This was considered as 100% coverage instead of the value above 100%. To evaluate the data from 2007, it was assumed that the unvaccinated individuals became vaccinated the following years. This resulted in 87% %COV and 4,961,868 unvaccinated individuals.

Table 5 shows the %COV value vs. the coverage as reported by PAHO and IMSS. An important difference is observed for the 2017–2020 period when the coverage reported by PAHO was compared to our calculated value. Furthermore, there is a considerable difference between the coverage values reported by IMSS and our own values.

Finally, Figure 4 shows the number of newly reported cases of respiratory and meningeal tuberculosis along with their respective %COV values.

## 4. Discussion

Based on the number of applied doses and the theoretical target population, i.e., births per year in the case of the BCG vaccine (2007–2021), our calculations show that between 3,917,616 and 4,961,868 individuals did not receive their vaccine dose.

The coverage %COV value we calculated differs regarding that reported by the respective institutions and by PAHO. A similar case was observed when the data from 2017 and 2020 were compared. Such coverage is not adequate to protect the population and consequently to prevent the disease. These values were less than 90% and 21% in 2020. Additionally, a lower number of vaccines was acquired between 2019 and 2021. For all institutions, there is a large difference between the calculated and the reported coverage values. This may be explained by the fact that IMSS and ISSSTE considered a much smaller target population when compared to the sized used for our calculations, whereas SSA considered a much larger population. Alternatively, it is possible that IMSS and ISSTE did not acquired the required number of doses. Conversely, SSA may have acquired and applied a larger amount to compensate for a vaccine shortage.

We queried different sources to obtain information regarding the acquisition of BCG vaccines during the 2006–2021 period. We obtained 87.5% of the amounts information of vaccine purchased by SSA and ISSSTE and 56% from IMSS.

Theoretically, the variation of the acquired amounts should be minimal because the target population is usually constant each year. However, we observed decreases and increases that have no logical explanation. Based on our data, it is not possible to identify the underlying cause of these variations for all vaccines. Moreover, we observed that vaccine acquisition dramatically decreased during the last three years and there is no explanation for this phenomenon.

Acquisitions regarding the theoretical target population and the percentage of applied vaccines were also reported. Based on this, a surplus of acquired vaccines was identified for the target population between 2006 and 2018. In some cases, 60% or less of such vaccines were applied. This results in a high wastage. We calculated that 46% of the acquired vaccine doses were wasted between 2006 and 2020. This may be explained using a 10-dose presentation of the vaccine.

Acquisition and coverage values are very different among institutions. In SSA, the number of acquired and applied vaccines is higher in comparison to the theoretical target population. In IMSS the number of acquired vaccines surpasses the target population but a small proportion is applied. Finally, ISSSTE acquires and applies an amount lower than the theoretical target population. This difference may be caused by the compensation made by one of the institutions in order to maintain the coverage in face of the failure shown by the other two.

Particularly, we observed that the number of applied vaccine doses surpassed the acquired amount in ISSSTE. In some cases, this may be explained by the existence of surpluses from previous years, but in other cases an explanation is lacking. This same phenomenon occurred in SSA in 2021. Similarly, there is no explanation with the remaining doses from the previous year.

We also identified reporting problems and poor data quality as there are inconsistencies among the same information retrieved from different sources.

The information is incomplete and it is not clear if this is because no vaccines were acquired in that particular year or because the information is not available. This is not clarified in any of the sources.

It is probable that some data are incorrect or unrealistic as they produced illogical results when they were pooled and evaluated. An example of this is the increased acquisitions by ISSSTE during the 2008–2011 period. It is possible that an information error occurred as the use of 10-dose bottles was reported instead of single doses. However, it is not possible to verify this and the data were used as expressed in the information source.

Finally, we identified that the number of newly reported cases of tuberculosis increased when coverage decreased.

Altogether, the data indicate that the scope of the vaccination programs has gradually decreased, and this situation may have been aggravated during the COVID-19 pandemic. The BCG vaccine is a representative example. This vaccine is applied right after birth under a single-dose scheme. When compared to other vaccines applied at older ages under multi-dose schemes, the problem probably may be even greater. It is necessary to identify the possible underlying causes of this issue. This work identifies some of them, such as different target populations considered by the respective institutions causing differences regarding acquisition and application when compared to the expected values. Vaccine purchase is not constant, and this may be caused by poor planning, shortages or low budget. Additionally, the program may not be under and adequate control. If this situation persists and if such course does not change, the incidence of vaccine-preventable diseases will further increase because of a decreased coverage.

Individuals have not an adequate access to the BCG vaccine because most of the major public institutions do not have a sufficient vaccine supply to provide coverage to their eligible population. Moreover, it is probable that beneficiaries do not take their infants to receive neonatal care or they are not aware of such benefit. As indicated by the data in this report, it is probable that many newborns did not receive the vaccine and they are still unprotected. Additional studies need to be conducted out to integrate the data obtained from all public and private institutions.

## 5. Conclusions

Currently, there are still individuals that have not received the BCG vaccine and remain unprotected. Application rates are less than 90% in some years. We estimated that between 3,917,616 and 4,961,868 individuals did not receive their BCG vaccine dose during the 2006–2021 period. It is necessary to establish vaccination campaigns in the country to cover those individuals that were not been vaccinated in previous years due to a vaccine shortage.

The access to the BCG vaccine is difficult in Mexico. The number of acquired vaccine doses were not sufficient for the target population in some years.

There is no explanation for the variable amount of acquired vaccines.

Approximately, 54% of the acquired vaccines doses were applied, thus a high wastage occurs. This may be prevented by ensuring the supply of a 5-dose presentation as it is registered in Mexico. This would reduce wastage by 50% and it may result in a decrease regarding the acquisition of vaccine doses by approximately the same percentage.

The vaccination coverage calculated in this study differs from that reported by the PAHO. Such differences are between 1% and 55%. We propose that a system controlling the application of vaccines must be established and it should be submitted to constant review. An acquisition control system is also necessary. It should contain data on the amounts and the nominal information of the target population. This system must be audited and reviewed annually. In addition, all government information on vaccines should be public and easily accessible and it should contain data on acquisitions, coverage, rejections, shortages and more.

## Figures and Tables

**Figure 1 vaccines-11-00337-f001:**
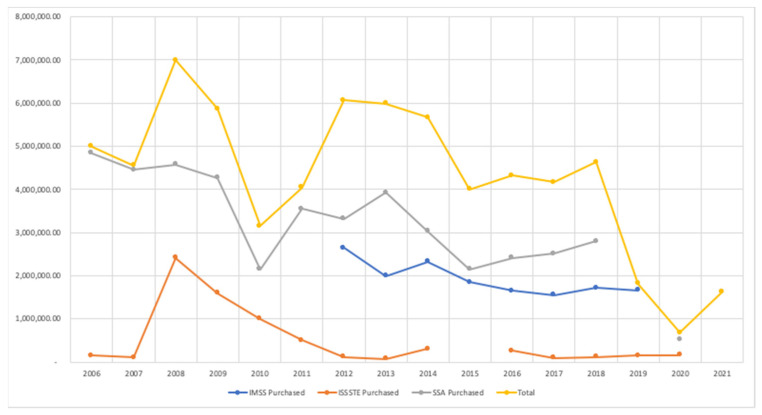
BCG vaccines. Annual acquisition.

**Figure 2 vaccines-11-00337-f002:**
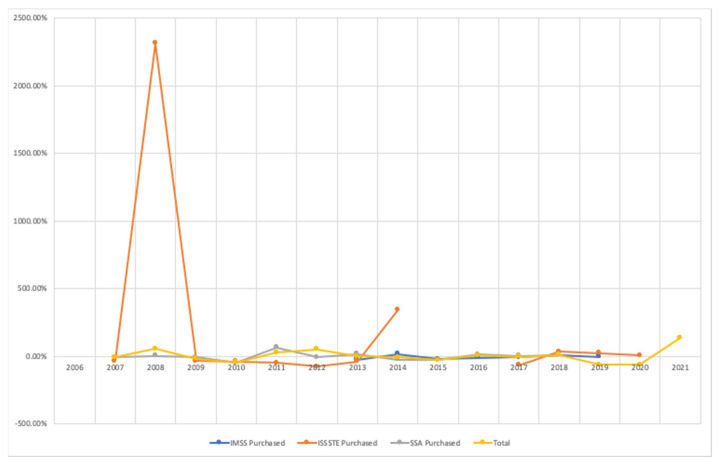
BCG vaccines. Percent variation of annual acquisition.

**Figure 3 vaccines-11-00337-f003:**
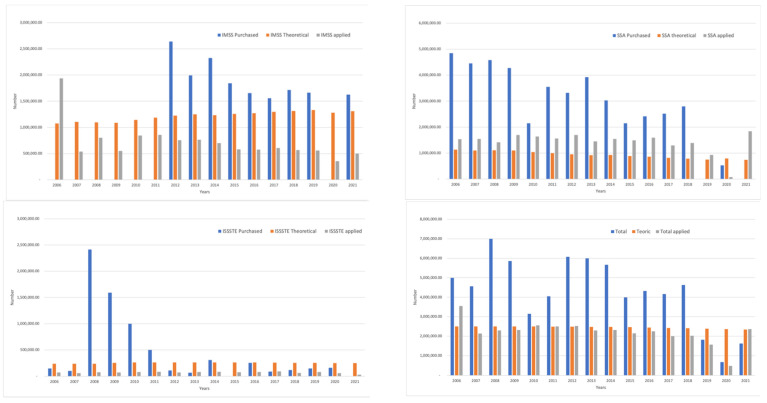
Comparison between total acquired BCG vaccines, theoretical target population and application as reported by institution.

**Figure 4 vaccines-11-00337-f004:**
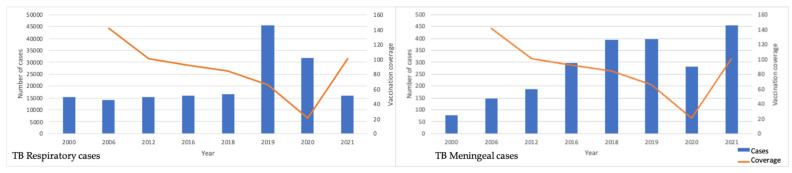
Number of newly reported cases of respiratory and meningeal tuberculosis compared to vaccine coverage.

**Table 1 vaccines-11-00337-t001:** Newly diagnosed cases of tuberculosis in Mexico: 2000–2021 period.

Disease/Year	2000 [6]	2006 [7]	2012 [8]	2016 [9]	2018 [10]	2019 [11]	2020 [12]	2021 [13]
Respiratory tuberculosis	15,201	13,985	15,334	16,082	16,700	45,637	31,724	16,008
Meningeal tuberculosis	78	146	187	297	395	396	282	455

**Table 2 vaccines-11-00337-t002:** Percentage of the population per institution per year (%).

	2006	2007	2008	2009	2010	2011	2012	2013	2014	2015	2016	2017	2018	2019	2020	2021
IMSS	43	44	44	44	46	47	49	50	50	51	52	54	55	56	54	56
ISSSTE	10	10	10	10	10	11	11	11	11	11	11	11	11	11	11	11
Other	2	2	2	2	2	2	2	2	1	2	2	1	1	2	2	2
SSA	45	44	44	44	42	40	38	37	38	36	35	34	33	31	33	31

**Table 3 vaccines-11-00337-t003:** Reported acquisition percentage.

Inst.	IMSS			ISSSTE			SSA			TOTAL		
Year	%PUR	%APP	%COV	%PUR	%APP	%COV	%PUR	%APP	%COV	%PUR	%APP	%COV
2006	ND	ND	180%	63%	47%	30%	426%	32%	135%	200%	71%	142%
2007	ND	ND	49%	42%	59%	25%	404%	35%	140%	183%	47%	86%
2008	ND	ND	73%	1010%	3%	32%	412%	31%	128%	280%	33%	92%
2009	ND	ND	51%	620%	5%	29%	387%	40%	154%	235%	40%	93%
2010	ND	ND	74%	381%	8%	31%	206%	76%	157%	126%	81%	103%
2011	ND	ND	72%	190%	17%	33%	356%	44%	157%	162%	62%	100%
2012	216%	29%	62%	42%	65%	27%	348%	51%	178%	244%	42%	101%
2013	160%	38%	61%	26%	118%	31%	427%	37%	158%	241%	38%	92%
2014	189%	30%	57%	116%	27%	32%	325%	51%	165%	229%	41%	94%
2015	147%	31%	46%	ND	ND	29%	242%	69%	168%	162%	54%	87%
2016	130%	35%	45%	97%	32%	31%	279%	66%	185%	177%	52%	92%
2017	120%	39%	47%	35%	102%	35%	307%	52%	158%	172%	48%	82%
2018	130%	33%	43%	47%	53%	24%	357%	50%	177%	193%	44%	84%
2019	125%	34%	42%	59%	54%	32%	ND	ND	124%	76%	87%	66%
2020	ND	ND	28%	64%	38%	24%	66%	14%	9%	29%	71%	21%
2021	124%	31%	38%	ND	ND	12%	ND	ND	249%	69%	146%	101%

PUR: purchase; App: application, COV: coverage.

**Table 4 vaccines-11-00337-t004:** BCG Theoretical target population, unvaccinated individuals and %COV.

	IMSS			ISSSTE		
Year	Theoretical Target Population	Unvaccinated (-)	%COV	Theoretical Target Population	Unvaccinated (-)	%COV
2006	1,075,406	861,654	180%	237,487	–166,865	30%
2007	1,106,289	–567,001	49%	238,172	–179,346	25%
2008	1,097,306	–297,522	73%	238,981	–162,692	32%
2009	1,087,164	–536,075	51%	256,433	–182,671	29%
2010	1,143,345	–297,434	74%	262,139	–181,229	31%
2011	1,184,657	–326,723	72%	263,372	–177,071	33%
2012	1,223,670	–468,210	62%	265,054	–192,404	27%
2013	1,248,826	–486,561	61%	265,045	–182,295	31%
2014	1,230,721	–530,741	57%	264,896	–180,362	32%
2015	1,258,834	–678,341	46%	263,991	–188,315	29%
2016	1,270,175	–694,654	45%	262,381	–180,670	31%
2017	1,298,020	–691,459	47%	260,374	–169,005	35%
2018	1,314,563	–746,055	43%	257,429	–194,387	24%
2019	1,332,778	–773,631	42%	255,381	–174,412	32%
2020	1,278,459	–925,914	28%	251,665	–191,597	24%
2021	1,310,312	–810,173	38%	250,441	–221,511	12%
Total	19,460,523	–7,968,838	59%	4,093,242	–2,924,833	29%
	**SSA**			**TOTAL**		
**Year**	**Theoretical** **Target** **Population**	**Unvaccinated (-)**	**%COV**	**Theoretical** **Target** **Population**	**Unvaccinated (-)**	**%COV**
2006	1,136,975	399,460	135%	2,499,866	1,044,252	142%
2007	1,102,122	441,254	140%	2,496,514	–355,024	86%
2008	1,110,804	307,409	128%	2,497,031	–202,745	92%
2009	1,103,480	590,794	154%	2,497,018	–177,893	93%
2010	1,041,861	592,861	157%	2,497,290	64,253	103%
2011	998,018	564,719	157%	2,495,966	11,006	100%
2012	953,526	746,073	178%	2,492,092	35,617	101%
2013	920,894	529,743	158%	2,484,455	–188,803	92%
2014	931,578	608,067	165%	2,476,730	–152,571	94%
2015	890,168	603,338	168%	2,462,237	–312,562	87%
2016	865,078	732,709	185%	2,446,565	–191,546	92%
2017	819,124	478,756	158%	2,426,039	–430,229	82%
2018	785,693	601,695	177%	2,405,801	–386,863	84%
2019	750,075	180,407	124%	2,385,953	–815,355	66%
2020	789,040	–715,978	9%	2,366,494	–1,880,819	21%
2021	739,646	1,100,296	249%	2,347,346	21,665	101%
Total	14,938,082	7,761,603	152%	39,277,395	–3,917,616	90%

**Table 5 vaccines-11-00337-t005:** Coverage: comparison with total values and those reported by IMSS.

	TOTAL		IMSS	
Year	Calculated	PAHO	Calculated	Reported
2006	142%	100%	180%	ND
2007	86%	100%	49%	ND
2008	92%	100%	73%	ND
2009	93%	99%	51%	ND
2010	103%	98%	74%	ND
2011	100%	99%	72%	95.2%
2012	101%	99%	62%	94.3%
2013	92%	91%	61%	97.1%
2014	94%	96%	57%	96.8%
2015	87%	100%	46%	101.2%
2016	92%	100%	45%	100.1%
2017	82%	94%	47%	107.5%
2018	84%	96%	43%	100.8%
2019	66%	76%	42%	102.4%
2020	21%	33%	28%	64.8%
2021	101%	100%	38%	ND

## Data Availability

All data are available in databases from the respective institution as cited in the references.

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
