# Peer review of "Report of 16 Years of the BCG Vaccine under the Expanded Program on Immunizations in Mexico (2006–2021)"

_vaccines, 2023, doi:10.3390/vaccines11020337_

Round 1
Reviewer 1 Report
Review of Report of 16 years of the BCG vaccine in the Expanded Program on Immunisations in Mexico (2006-2021)
Rodrigo Romero-Feregrino 1,2,*; Raul Romero-Cabello 3,4; Mario Alfredo Rodríguez-León 5; Raul Romero-Feregrino 6,7; Berenice Muñoz-Cordero 8; Julieta Isabel Aguilar-Feregrino 9
1. There are is one possible example of plagiarism in this paper: on page 2, paragraph five reads “BCG vaccination of infants, at birth or as soon as possible after birth, is one of the key components of pillar 1 of the End TB Strategy. It has been estimated that high global cov-erage (90%) and widespread use of BCG in routine infant vaccination programs could prevent over 115,000 TB deaths per birth cohort in the first 15 years of life.”
This is a direct lift from the IAP Guidebook on Immunization 2018-2019 page 95, first paragraph, which read “Bacillus Calmette–Guérin (BCG) vaccination of infants, at birth or as soon as possible after birth, is one of the key components of pillar 1 of the End-TB Strategy. It has been estimated that high global coverage (90%) and widespread use of BCG in routine infant vaccination programs could prevent over 115,000 TB deaths per birth cohort in the first 15 years of life.”
It is not cited.
Balasubramanian S, Digant D Shastri, Pallab Chatterjee, Abhay K Shah, Harish K Pemde, Shivananda S, Vijay Kumar Guduru IAP Guidebook on Immunization 2018-2019
Jaypee Brothers Medical Publishers, Nov 28, 2019 – Medical
2. Why has the scope of the Universal Vaccination Program decreased? I think discussing this would make the article more interesting.
3. Why do people have difficulty in accessing vaccination? Again, I think this would increase interest in the article.
Author Response
Point 1: There are is one possible example of plagiarism in this paper: on page 2, paragraph five reads “BCG vaccination of infants, at birth or as soon as possible after birth, is one of the key components of pillar 1 of the End TB Strategy. It has been estimated that high global cov-erage (90%) and widespread use of BCG in routine infant vaccination programs could prevent over 115,000 TB deaths per birth cohort in the first 15 years of life.”
Response 1: We take that paragraph from reference 1, which is found on page 83 paragraph 2, together with the next paragraph in the work that is found on page 92 paragraph 3, for that reason both are referenced with the number 1, if necessary we can put reference 1 in both paragraph to avoid confusion. Reference 1 is World Health Organization. BCG vaccine: WHO position paper, February 2018 - Recommendations. Vaccine. 2018 Jun 7;36(24):3408-3410. doi: 10.1016/j.vaccine.2018.03.009. Epub 2018 Mar 30. PMID: 29609965.
Point 2: Why has the scope of the Universal Vaccination Program decreased? I think discussing this would make the article more interesting.
Response 2: Thank you very much for the proposal, which we think is excellent, we would add the following paragraph in the discussion:
All these data show us that the scope of the vaccination program has gradually decreased, which became more acute during the COVID-19 pandemic. We exemplify this with a BCG vaccine, which is applied at birth and that the scheme is one dose, if we compare it with vaccines that are applied at other ages and have schemes of more than a few doses, it is likely that the problem is greater. Possible causes for this problem should be sought, with this work we can identify some such as a probable difference in the target population between institutions and therefore a different purchase and application than expected, purchases are not constant which can be due to planning, shortages or budget, and there is a probable disorganization in the control of the program. If this persists and the course is not changed, it is likely that a further increase in vaccine-preventable diseases will be observed due to the decrease in coverage.
Point 3: Why do people have difficulty in accessing vaccination? Again, I think this would increase interest in the article.
Response 3: We would add the following paragraph in the discussion:
People have difficulty accessing the BCG vaccine in their different institutions, because the most important public institutions do not have the amount of vaccines to apply to all their eligible population, it is also probably that people do not go to receive birth medical care in public institutions or do not know where they have the right to receive it, but with the data available for this report, many newborns probably did not have access to the vaccine and are unprotected, although other studies would need to be carried out integrating all the public and private institutions.
Reviewer 2 Report
This paper analyses health and cost implications of Bacillus Calmette and Güérin vaccine in Mexico over a 16-year period. The valuable effort is appreciable, the statistical analysis is adequate, and the structure of the text is satisfactory. However, there are some points in the text rising concerns. In many parts the analysis should be extended and the text must be rewritten:
1) This reviewer recommends a general revision of the text by a native English speaker or other qualified subjects, in order to evaluate the appropriateness of grammar and syntactic choices.
2) The acronym BCG should be explained in the abstract.
3) Figures: all characters should be enlarged, in the current form figures are impossible to read. Figure 4 needs to be re-formatted.
4) Page 1 “The Bacillus Calmette and Güérin (BCG) vaccines are among the oldest vaccines and were first used in humans in 1921. BCG is a live attenuated bacterial vaccine derived from Mycobacterium bovis that was originally isolated in 1902 from a tuberculous cow. WHO recommends that all BCG vaccines used in immunisation programs adhere to WHO standards. BCG is currently the only available TB vaccine.” References are missing. Also, please kindly explain timing and doses of vaccination, and how many times BCG vaccine is given to complete the vaccination process, in order to understand if this may represent a factor discouraging the target population. The target population for receiving BCG vaccine is not indicated in the text.
5) Page 1 “While BCG has demonstrated significant effectiveness in several populations, protection has not been consistent against all forms of TB and in all age groups.” Please kindly expand the concept, explaining the current knowledge about the mentioned variability. Also, please cite adequate references, and briefly discuss the alternatives that are currently being evaluated to replace BCG vaccine and to overcome BCG limitations. Otherwise, please briefly discuss how these alternatives would be beneficial in increasing TB prevention in the “Discussion” section, in the light of the experimental evidences reported in this paper.
6) Page 2 “In Mexico the BCG vaccine was first applied as of 1951, although it was not until 1991 when this and other vaccines were applied to children ≤ 5 years of age through government planning and inter-institutional logistics called "Universal Vaccination Program", with which the eradication of one disease, the control of others and vaccination coverage ≥ 90% were achieved”. Please kindly mention what diseases were eradicated and controlled through the proposed vaccines, and why this piece of information is important for the paper. Also please mention the reached coverage for BCG vaccine, and the role of the Universal Vaccination Program in helping to reach a higher BCG vaccine coverage.
7) Page 2 “However, in recent years this scope has decreased, and its consequence is reflected in the increase in some vaccine-preventable diseases. The case of tuberculosis stands out, which as a primary infection or facilitated by immunocompromised diseases has re-emerged as a public health problem.”. Please kindly cite adequate references.
8) Page 2 “BCG vaccination of infants, at birth or as soon as possible after birth, is one of the key components of pillar 1 of the End TB Strategy.” Please kindly explain what End TB Strategy is, and add supporting references.
9) Page 2 “It has been estimated that high global coverage (90%) and widespread use of BCG in routine infant vaccination programs could prevent over 115,000 TB deaths per birth cohort in the first 15 years of life.” Please add references.
10) Page 2 “Mexico has a long history in the application and production of vaccines, having had great successes and considerable achievements. Beginning in 1804 with the introduction of the smallpox vaccine, which was decreed mandatory in the year 1926.”. Syntax has no sense.
11) Page 2 “There are some general questions: what can be attributed to this situation that decreases vaccination coverage? What are the causes? And, how can we improve the situation?”. This reviewer warmly suggests to eliminate this part since a comprehensive evaluation of all (including hygienic) causes that may lead to an increase in TB incidence is missing in this paper. Also, the “situation” mentioned in the first question is not clear. Do authors mean the increase in TB cases, the reduction in BCG vaccine coverage, or both?
12) Page 3 “The types of data necessary to perform the study were defined”. Please kind list the types of collected data, or add some sentence like “please see below”.
13) Page 3 “Theoretical target population: It is the number of people who should be vaccinated according to the population of each institution and the vaccine indication”. Please kindly mention what source (WHO guidelines, rather than National programmes for example) has been used to determine vaccine indication. Also, please kindly explain for what categories of subjects BCG vaccination is indicated.
14) “Results” section: a number of aspects should be deepened in order to explore the causes leading to a drop in BCG vaccine administration rates. What is the target pool (new-borns, children, healthcare workers, other categories)? Data should be stratified according to the categories of subjects for which the administration of BCG vaccine is indicated. For example, what are the characteristics of patients that received BCG vaccine according to the indications? Are these subjects for example paediatric rather than adult subjects? Do these data take into account BCG vaccine administration to healthcare workers, or only new-borns/children are considered to extrapolate the data? Is there a specific category of subjects in which cases lacking BCG vaccine tend to accumulate? Is there any correlation with the socio-economic status of recipients? In the Institutions where the amount of purchased vaccines is smaller than the actual target population, are there any critical issues that may explain such a discrepancy? Do the three mentioned Institutions (IMSS, SSA and ISSSTE) distribute vaccines at all levels throughout the territory? What kind of centres administer BCG vaccines to the target population on the territory? Are these centres capillary distributed and easily reachable on the whole national territory? Are there zones where BCG vaccines are not distributed or are unavailable?
15) “Discussion” section: a number of aspects should be explained to help the readers in the understanding of data. Do target subjects/parents receive clear and adequate information for example at school or by family physicians? Could the discrepancy between the amount of purchased vaccines and the target population be explained in terms financial choices performed by the local Government unmatching scientific needs/evidences? Was the budget destined for the purchase of vaccines kept constant throughout all the analysed time frame?
16) Page 8 “Another possible explanation could be that IMSS and ISSTE do not buy enough and SSA buys and applies more to compensate for that lack of vaccines.” and Page 9 “In SSA we see that the number of vaccines bought and applied is higher than the theoretical target population. In IMSS the number of vaccines bought is higher than the target population but fewer are applied, and in ISSSTE fewer are purchased and ap-plied than the theoretical target population. This can be explained by the fact that one institution compensates for the failure of the others to maintain the total coverage.” Please kindly explain why it is necessary to reiterate the concept.
17) Page 9 “There are many data that are likely incorrect or not real as when they are evaluated together, they are not logical, for example, we were highlighted by the increase of purchased in the ISSSTE from 2008 to 2011, we believe that it may be an error in the information where bottles with 10 doses are reported instead of single doses, but since we do not know it, we record it as it is expressed in the information source.” According to these concerns, please kindly state the value of this paper, given that (by authors' admission) some data are not reliable. If the source of error could not be eliminated, the Authors should consider to avoid publishing this paper (at least in the present form), or to exclude the part of the paper affected by the errors.
Author Response
Point 1: This reviewer recommends a general revision of the text by a native English speaker or other qualified subjects, in order to evaluate the appropriateness of grammar and syntactic choices.
Response 1: Yes, we will send it back for a language review with the final version.
Point 2: The acronym BCG should be explained in the abstract.
Response 2: we will add Bacillus Calmette and Güérin (BCG) in the abstract
Point 3 Figures: all characters should be enlarged, in the current form figures are impossible to read. Figure 4 needs to be re-formatted.
Response 3: In the version with the format of the Vaccines magazine, figure 4 was messed up, we will send it in another format to avoid errors and we will send some higher quality images so that
Point 4 Page 1 “The Bacillus Calmette and Güérin (BCG) vaccines are among the oldest vaccines and were first used in humans in 1921. BCG is a live attenuated bacterial vaccine derived from Mycobacterium bovis that was originally isolated in 1902 from a tuberculous cow. WHO recommends that all BCG vaccines used in immunisation programs adhere to WHO standards. BCG is currently the only available TB vaccine.” References are missing. Also, please kindly explain timing and doses of vaccination, and how many times BCG vaccine is given to complete the vaccination process, in order to understand if this may represent a factor discouraging the target population. The target population for receiving BCG vaccine is not indicated in the text.
Response 4: The reference is the same as the following paragraph, the number 1, We can add the number in superscript in both paragraphs if necessary. The explanation of timing, doses and target population is in the following paragraph: …a single dose of BCG vaccine should be given to all healthy neonates at birth, for pre-vention of TB and leprosy. If the BCG vaccine cannot be given at birth, it should be given at the earliest opportunity thereafter and should not be delayed, to protect the child before exposure to infection occurs
Point 5: Page 1 “While BCG has demonstrated significant effectiveness in several populations, protection has not been consistent against all forms of TB and in all age groups.” Please kindly expand the concept, explaining the current knowledge about the mentioned variability. Also, please cite adequate references, and briefly discuss the alternatives that are currently being evaluated to replace BCG vaccine and to overcome BCG limitations. Otherwise, please briefly discuss how these alternatives would be beneficial in increasing TB prevention in the “Discussion” section, in the light of the experimental evidences reported in this paper.
Response 5: We can add the next text: While BCG has demonstrated significant effectiveness in several populations, protection has not been consistent against all forms of TB and in all age groups, with a range of 44%-99% of protection for pulmonary tuberculosis in neonates, 85%-90% for severe TB, 73% for TB meningitis and 77% miliary TB. Therefore, several new TB candidate
vaccines are in development, some of which are in advanced clinical trials. Some are designed to be used for booster vaccination following neonatal BCG vaccination.1
Point 6: Page 2 “In Mexico the BCG vaccine was first applied as of 1951, although it was not until 1991 when this and other vaccines were applied to children ≤ 5 years of age through government planning and inter-institutional logistics called "Universal Vaccination Program", with which the eradication of one disease, the control of others and vaccination coverage ≥ 90% were achieved”. Please kindly mention what diseases were eradicated and controlled through the proposed vaccines, and why this piece of information is important for the paper. Also please mention the reached coverage for BCG vaccine, and the role of the Universal Vaccination Program in helping to reach a higher BCG vaccine coverage.
Response 6: The commented text is not necessary for the article, the objective was to explain that the vaccination program in Mexico has been running for several years with successes such as the eradication of smallpox, and the control of polio, tetanus, rubella and measles. We can delete this text without problem.
Point 7: Page 2 “However, in recent years this scope has decreased, and its consequence is reflected in the increase in some vaccine-preventable diseases. The case of tuberculosis stands out, which as a primary infection or facilitated by immunocompromised diseases has re-emerged as a public health problem.”. Please kindly cite adequate references.
Response 7: Without problem we put the appropriate reference.
Point 8: Page 2 “BCG vaccination of infants, at birth or as soon as possible after birth, is one of the key components of pillar 1 of the End TB Strategy.” Please kindly explain what End TB Strategy is, and add supporting references.
Response 8: We add the following in the paragraph: BCG vaccination of infants, at birth or as soon as possible after birth, is one of the key components of pillar 1 of the End TB Strategy. It is a WHO program with the aims to reduce the number of TB deaths by 95% by 2035 compared to the 2015 rates.
Point 9: Page 2 “It has been estimated that high global coverage (90%) and widespread use of BCG in routine infant vaccination programs could prevent over 115,000 TB deaths per birth cohort in the first 15 years of life.” Please add references.
Response 9: Without problem we put the appropriate reference, is 1.
Point 10: Page 2 “Mexico has a long history in the application and production of vaccines, having had great successes and considerable achievements. Beginning in 1804 with the introduction of the smallpox vaccine, which was decreed mandatory in the year 1926.”. Syntax has no sense.
Response 10: okay. We would modify the syntax with the revision that would be made to the language.
Point 11: Page 2 “There are some general questions: what can be attributed to this situation that decreases vaccination coverage? What are the causes? And, how can we improve the situation?”. This reviewer warmly suggests to eliminate this part since a comprehensive evaluation of all (including hygienic) causes that may lead to an increase in TB incidence is missing in this paper. Also, the “situation” mentioned in the first question is not clear. Do authors mean the increase in TB cases, the reduction in BCG vaccine coverage, or both?
Response 11: The objective of these questions is to provide the reader with the doubts that as authors arise about the situation described in the introduction, the questions refer to the coverage as the first question says, in addition, at suggestion of another reviewer, a hypothesis of the work will be added, which is the next: In this report, we expect to find a low concordance between the number of vaccines purchased, applied, and coverage, with a large difference between the data we calculated and those obtained from the institutions studied.
Point 12: Page 3 “The types of data necessary to perform the study were defined”. Please kind list the types of collected data, or add some sentence like “please see below”.
Response 12: We propose to modify the paragraph: The types of data necessary to perform the study were beneficiaries by institution, description of the vaccine, number of doses acquired, and coverage from ISSSTE, IMSS and Ministry of Health (SSA), and coverage reported to the Pan-American Health Organization (PAHO). The data was taken from Compranet21, IMSS purchase22 and PAHO23.
Point 13: Page 3 “Theoretical target population: It is the number of people who should be vaccinated according to the population of each institution and the vaccine indication”. Please kindly mention what source (WHO guidelines, rather than National programmes for example) has been used to determine vaccine indication. Also, please kindly explain for what categories of subjects BCG vaccination is indicated.
Response 13: The indications are those defined by the WHO in its position paper (reference one), which is found in the article on page 1, second paragraph where it says:
In countries or settings with a high incidence of TB and/or high leprosy burden, a single dose of BCG vaccine should be given to all healthy neonates at birth, for preven-tion of TB and leprosy. If the BCG vaccine cannot be given at birth, it should be given at the earliest opportunity thereafter and should not be delayed, to protect the child be-fore exposure to infection occurs
Point 14: “Results” section: a number of aspects should be deepened in order to explore the causes leading to a drop in BCG vaccine administration rates. What is the target pool (new-borns, children, healthcare workers, other categories)? Data should be stratified according to the categories of subjects for which the administration of BCG vaccine is indicated. For example, what are the characteristics of patients that received BCG vaccine according to the indications? Are these subjects for example paediatric rather than adult subjects? Do these data take into account BCG vaccine administration to healthcare workers, or only new-borns/children are considered to extrapolate the data? Is there a specific category of subjects in which cases lacking BCG vaccine tend to accumulate? Is there any correlation with the socio-economic status of recipients? In the Institutions where the amount of purchased vaccines is smaller than the actual target population, are there any critical issues that may explain such a discrepancy? Do the three mentioned Institutions (IMSS, SSA and ISSSTE) distribute vaccines at all levels throughout the territory? What kind of centres administer BCG vaccines to the target population on the territory? Are these centres capillary distributed and easily reachable on the whole national territory? Are there zones where BCG vaccines are not distributed or are unavailable?
Response 14: The target pool are newborn this indications are defined by the WHO in its position paper (reference one).
The institutions that were studied are found throughout all the country with the 3 levels of care, there are data that we do not have to answer the questions or that cannot be answered with the data obtained.
Point 15: “Discussion” section: a number of aspects should be explained to help the readers in the understanding of data. Do target subjects/parents receive clear and adequate information for example at school or by family physicians? Could the discrepancy between the amount of purchased vaccines and the target population be explained in terms financial choices performed by the local Government unmatching scientific needs/evidences? Was the budget destined for the purchase of vaccines kept constant throughout all the analysed time frame?
Response 15: The BCG vaccine is indicated for its application to newborns before they leave the hospitals, that is why it is a vaccine that exemplifies a serious situation in the vaccination program.
Regarding the other questions, we do not have an answer, becase we don´t have the budget data for all these years, but it would be very interesting to do it in a future work, adding the budget and sub-exercise of this budget.
Point 16: Page 8 “Another possible explanation could be that IMSS and ISSTE do not buy enough and SSA buys and applies more to compensate for that lack of vaccines.” and Page 9 “In SSA we see that the number of vaccines bought and applied is higher than the theoretical target population. In IMSS the number of vaccines bought is higher than the target population but fewer are applied, and in ISSSTE fewer are purchased and ap-plied than the theoretical target population. This can be explained by the fact that one institution compensates for the failure of the others to maintain the total coverage.” Please kindly explain why it is necessary to reiterate the concept.
Response 16: Okay, we propose to be less repetitive to delete the last sentence that says:
This can be explained by the fact that one institution compensates for the failure of the others to maintain the total coverage.
Point 17: Page 9 “There are many data that are likely incorrect or not real as when they are evaluated together, they are not logical, for example, we were highlighted by the increase of purchased in the ISSSTE from 2008 to 2011, we believe that it may be an error in the information where bottles with 10 doses are reported instead of single doses, but since we do not know it, we record it as it is expressed in the information source.” According to these concerns, please kindly state the value of this paper, given that (by authors' admission) some data are not reliable. If the source of error could not be eliminated, the Authors should consider to avoid publishing this paper (at least in the present form), or to exclude the part of the paper affected by the errors.
Response 17: We disagree with this comment, because part of this work and the evaluation of the vaccination programs is the adequate reporting of the data, to evaluate the success or the needs of the program and the population, we consider that we need to evaluated and discussed with the same data used by institutions to make decisions, in order to improve this and other problems that can be detected in the future.
For these reasons we commented in the discussion: “We also observed that there may be problems with reporting and data quality be-cause there are inconsistencies in the information found in different sources. There are confusing and different data depending on the source.
The data are incomplete and we cannot be sure if this is because the vaccines were not acquired that year or because the information is not available, and this is not indi-cated in the sources.”
For the same reason we recommend in the conclusion: “We propose to implement a system to control the application of vaccines, which is con-stantly reviewed, and a purchasing control system with the quantity data, with nomi-nal information for the target population, which can be audited and reviewed annual-ly”.
Reviewer 3 Report
The research article entitled: Report of 16 years of the BCG vaccine in the Expanded Program on Immunisations in Mexico (2006-2021) is very well presented.
Minor modifications are required.
In the abstract rewrite the concluding remarks.
In the last section of introduction emphasize your research hypothesis.
In discussion, mention about previous work like screening of for Tuberculosis Drug Candidates Utilizing a Luciferase-Expressing Recombinant Mycobacterium bovis Bacillus Calmette-Guéren
Add some information in details about the assays used.
Author Response
Point 1: In the abstract rewrite the concluding remarks.
Response 1: We propose the following change: There are low coverage and people unprotected by BCG vaccine, there were difficulty accessing the BCG vaccine and the number of new reported cases of tuberculosis increases in the last years. It is necessary to implement vaccination campaigns to protect the unvaccinated people and implement a nominal control system for coverage, purchases and target population.
Point 2: In the last section of introduction emphasize your research hypothesis.
Response 2: We propose to add the following paragraph: In this report, we expect to find a low concordance between the number of vaccines purchased, applied, and coverage, with a large difference between the data we calculated and those obtained from the institutions studied.
Point 3: In discussion, mention about previous work like screening of for Tuberculosis Drug Candidates Utilizing a Luciferase-Expressing Recombinant Mycobacterium bovis Bacillus Calmette-Guéren
Response 3: We believe that the suggested study is not related to what we did, because it is about Tuberculosis Drug Candidates, and our study is an analysis of data on the performance of the vaccination program, we did not test or search for any new TB drug.
Point 4: Add some information in details about the assays used.
Response 4: We do not use any assay or diagnostic tests, we only use data and perform mathematical formulas to analyze the data from the different institutions and the population of Mexico, which are referred to in the methodology.
Round 2
Reviewer 1 Report
The writing needs to be improved. For example, in the introduction on page 1 you have the sentence ". WHO recommends that any BCG vaccine used in immunization programs should comply with the their standards." vYou should drop "the".
Then You follow with a repeated sentence ". Currently, BCG is the only TB vaccine available. Currently, BCG is the only TB vaccine available." One of these needs to be eliminated.
The document needs to be proofread again before resubmission, please.
Author Response
We did a review again, and we corrected some points that we found, as well as those commented by you.
Reviewer 2 Report
This reviewer would like to thank the authors for their effort. Please kindly note that there is a sentence containing a repetition (page 1) “Currently, BCG is the only TB vaccine available. Currently, BCG is the only TB vaccine available.”.
Author Response
We did a review again, and we corrected some points that we found, as well as those commented by you.
Thank you very much.